# Post-surgery spontaneous pneumothorax: Long-term recurrence rates and follow-up challenges revealed by a written survey

Ryo Nonomura⬚*, Ryuga Yabe, Yutaka Oshima, Takanobu Sasaki, Naoya Ishibashi⬚, Takafumi Sugawara

Department of Thoracic Surgery, Tohoku Medical and Pharmaceutical University, Sendai City, Japan

* r.nono616@gmail.com

## Abstract

### Background

Spontaneous pneumothorax (SP) is a frequently encountered respiratory condition. Despite advancements in treatment techniques, there is currently no standardized treatment protocol. The aim of this study was to standardize the management of SP by collecting long-term postoperative data through written surveys.

### Methods

Our study included 673 surgeries performed for SP at our institution between January 2011 and December 2019. We administered written surveys via mail to gather data on post-surgery recurrence rates and other related factors. The survey addressed key symptoms, medical consultations, and specific diagnoses of SP after surgery.

### Results

The effective response rate was 70.7%, with significant differences observed across age groups. Among teenagers, the recurrence rate peaked three years postoperatively, with an ipsilateral recurrence rate of 4.8% and a contralateral incidence rate of 11.8%, both of which were significantly higher than those in other age groups. Additionally, age, sex, postoperative contralateral pneumothorax, and the period before the survey were identified as factors influencing the survey results.

### Conclusion

This survey highlighted the need for long-term postoperative follow-up for teenagers. While written surveys have limitations, this study provided valuable data for understanding the outcomes of SP. Moving forward, it is advisable to conduct surveys using other effective tools or to continue follow-ups in outpatient clinics.

**Data Availability Statement:** All relevant data are within the paper.

**Funding:** The author(s) received no specific funding for this work.

**Competing interests:** The authors have declared that no competing interests exist.

## Introduction

Spontaneous pneumothorax (SP) is a frequently encountered respiratory condition. However, owing to differences in treatment policy and follow-up period across facilities, there is a lack of treatment specific to the pathophysiology of the disease. The recurrence rate after conservative treatment with rest, simple aspiration, and drainage of the thoracic cavity is reported to be approximately 30%. Oftentimes SP is discovered incidentally during physical examinations, or it causes only minor symptoms and spontaneously resolves. On the other hand, surgery is recommended for the second or subsequent pneumothoraces based on the observation that 70% of spontaneous pneumothoraces do not recur [1, 2]. In Japan, approximately 14,000 cases of SP are treated surgically each year. Despite daily improvements in surgical techniques, the postoperative recurrence rate can reach 20% [3–5] among different centers, and the actual recurrence rate is still unknown. We believe that it is necessary to assess the accuracy of the recurrence rate after not only conservative treatment but also after surgical treatment to aid the development of guidelines for the treatment of SP in the future.

Although follow-up studies of SP have been conducted at many institutions, most cases of SP have been reported to recur within 1 year of initial onset [1], and the follow-up period is often as short as 1–2 years. The methods include written surveys by mail [5], telephone interviews [4, 6–10] and questionnaire surveys [6]. Since the enactment of the Personal Information Protection Law in Japan, there is a strong tendency to avoid telephone interview surveys without prior notification of personal health information. Harris et al. [11] 1997 reported that a telephone interview survey was more cost-effective than a mailed written survey because of the higher response rate. However, telephone surveys, in which the researcher interviews the subjects directly, do not yield accurate results if the subject experiences difficulty in answering [10].

In this study, as a high-volume center for SP treatment, we administered a written survey on thoracoscopic surgery by mail to determine the actual treatment status. We report the usefulness of this survey method and its issues on the basis of the follow-up rate.

## Materials and methods

The study was approved by the Clinical Research Review Committee, Tohoku Medical and Pharmaceutical University Hospital(IRB: 2020-2-036), and we sent the first questionnaire on July 20, 2020. Cases in which written informed consent was obtained at the time of the survey were included in the study. Only one patient was excluded from the study because consent could not be obtained.

Among the 983 patients who developed SP at our institution between January 2011 and December 2019, 704 involved patients between the ages of 10 and 60 years. Of these, 31 were cases whose progress could not be ascertained and whose deaths were confirmed. After these cases were excluded, 673 cases were included in the study, and the survey was conducted in by mail. The patients' mailing addresses were retrieved from the medical records at the time of surgical admission. A written survey form was enclosed with the documents confirming consent to this study and sent out.

The written survey form contained three questions, which were essentially yes/no questions. Question 1 asked whether the subjects were aware of any symptoms (chest pain, back pain, dyspnea) that might suggest the onset of pneumothorax from the time of surgery to the date of the survey. Question 2 asked whether the patient visited a medical institution after becoming aware of the symptoms. Question 3 asked whether the patient had been diagnosed with SP on examination and whether the pneumothorax was on the left side, right side, or bilateral.

The study revealed the following items: 1) effective reply rate (number of replies/number of no replies/number of unknown recipients), 2) age at the time of surgery, 3) sex (male/female), 4) smoking history at the time of surgery, 5) postoperative complications, 6) postoperative ipsilateral recurrence, 7) postoperative contralateral pneumothorax, 8) place of residence at the time of surgery (local/outside city), 9) hospital days after surgery, and 10) the period from surgery to the mailing date of the written survey. The time period was defined as the period from the time of surgery to the date the written survey was mailed.

## Statistical analysis

The mean and standard deviation were calculated for continuous variables, and the distribution of nominal variables was assessed for the three groups divided by whether a reply was received. One-way analysis of variance (ANOVA) was used for continuous variables, and Fisher's exact test was used for binary variables. Logistic regression analysis was employed to determine the significance of the effect on the response, and factors with a p value < 0.05 in the univariate analysis were considered significant.

## Results

### Effectiveness of the follow-up survey method via a written survey form

Fig 1 shows the number of replies, unanswered questions, and unknown recipients by age group, and Fig 2 shows the reply rate and the unknown recipient rate.

The actual reply rate was 55.9% (376 cases/673 cases), and the effective reply rate was 70.7% (376 cases/532 cases), excluding the 141 cases with unknown addresses.

The effective response rates by age group were 81.8% (153 cases/187 cases) for the 10 s, 70.9% (134 cases/189 cases) for the 20 s, 72.6% (77 cases/106 cases) for the 30 s, 89.1% (57

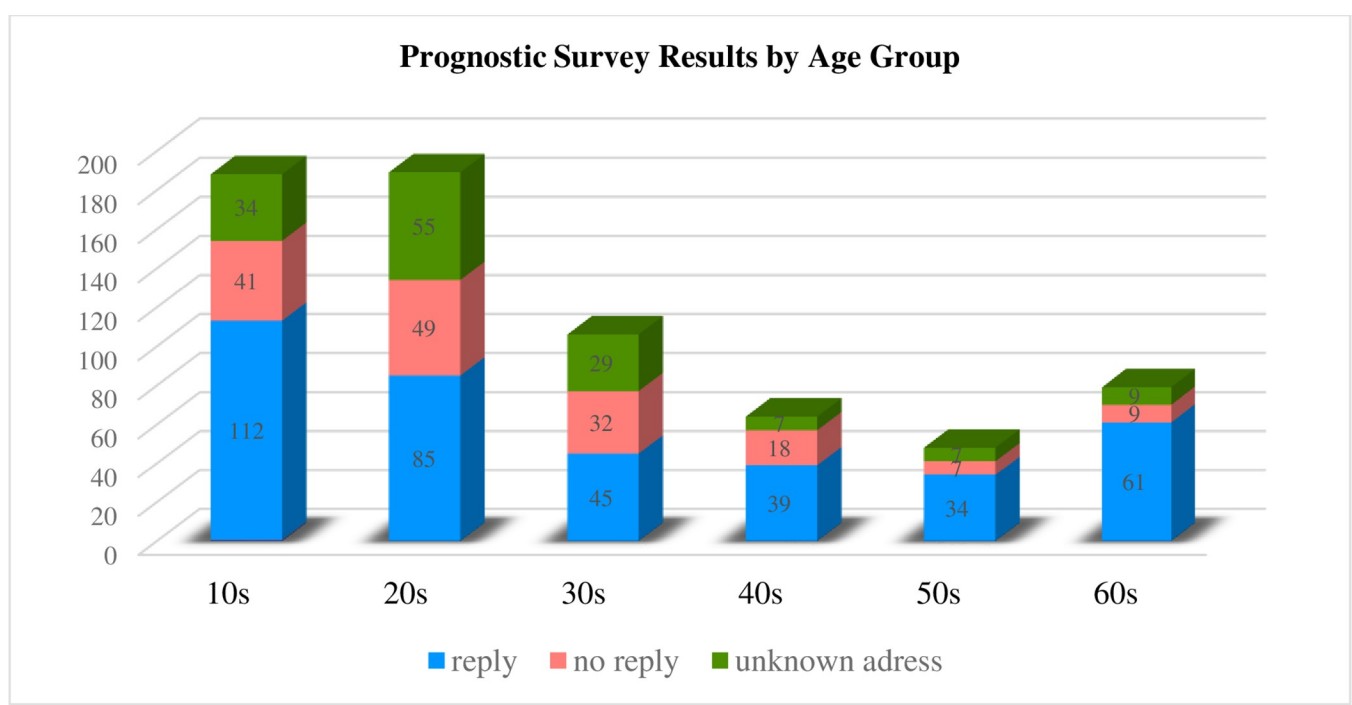

**Fig 1. Prognostic survey results by age group.** The number of replies, the number of unanswered replies, and the number of unknown recipients by year are shown in the bar graphs.

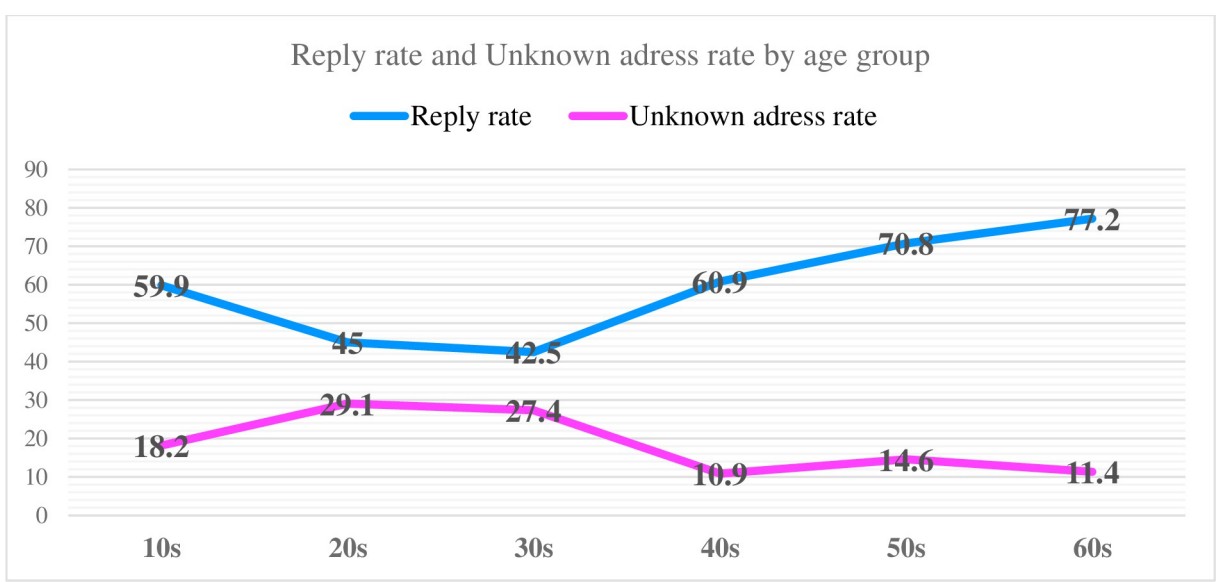

**Fig 2. Reply rate and unaddressed rate by age group.** Line graph showing the relationship between the percentage of replies and the percentage of unknown recipients by year and age.

cases/64 cases) for the 40 s, 85.4% (41 cases/48 cases) for the 50 s, and 88.6% (70 cases/79 cases) for the 60 s. (70 cases/79 cases), 60 s: 88.6% (70 cases/79 cases)

In terms of the effective reply rate, those in their 60s had significantly higher reply rates than did those in their 20s and 30 s (20 s vs. 60 s: P = 0.0057, 20 s vs. 60 s: P = 0.0017).

## Patient background contributing to effective replies

The three groups were divided into three groups (reply, no reply, and unknown address), and each observation item was compared (Table 1).

There were 376 patients in the reply group (328 males and 48 females). The mean age was 34.6±17.9 years. A total of 197 patients (52.4%) had a history of smoking. Surgery was performed due to the first occurrence of pneumothorax in 292 patients (77.7%), postoperative complications in 23 patients (6.1%), postoperative ipsilateral recurrence in 17 patients (4.5%),

**Table 1. Patient background by reply status.**

|  | reply | no reply | unknown | P value |
|---|---|---|---|---|
|  | (n = 376) | (n = 156) | (n = 141) |  |
| Age | 34.6±17.9 | 30.2±13.7 | 29.5±13.2 | 0.00083 |
| Sex(male or female) | 328/48 | 148/8 | 131/10 | 0.01 |
| Smoking history | 197 | 94 | 84 | 0.15 |
| Postoperative complications | 23 | 8 | 4 | 0.34 |
| Postoperative ipsilateral recurrence | 17 | 4 | 3 | 0.39 |
| Postoperative contralateral incidence | 31 | 2 | 1 | <0.0001 |
| Place of residence at the time of surgery (local or outside the city) | 182/194 | 71/85 | 94/47 | 0.0002 |
| Postoperative hospital days | 3.6±3.1 | 3.5±4.8 | 3.5±3.2 | 0.95 |
| Time from surgery to survey | 6.8±2.5 | 7.2±2.5 | 8.4±2.0 | <0.0001 |

Patients were grouped by survey completion: those who replied, those who did not reply, and unknown, and patient background data were compared between groups.

and postoperative contralateral occurrence in 31 patients (8.2%). A total of 182 patients (48.4%) lived in the city, and 194 (51.6%) lived outside the city. The mean postoperative drainage period was 1.5±2.0 days, and the mean postoperative hospital stay was 3.6±3.1 days. The mean elapsed time from surgery to the survey was 6.8 ± 2.5 years.

There were 156 patients in the no reply group, including 148 males and 8 females. The mean age was 30.2±13.7 years. The number of patients with a history of smoking was 94 (60.3%). Surgery was performed due to the first occurrence of pneumothorax in 142 patients (91.0%), postoperative complications in 8 patients (5.1%), postoperative ipsilateral recurrence in 4 patients (2.6%), and postoperative contralateral occurrence in 2 patients (1.3%). Seventy-one patients (45.5%) lived in the city, and 85 (54.5%) lived outside the city. The mean postoperative drainage period was 1.4±1.4 days, the mean postoperative hospital stay was 3.6±4.8 days, and the mean elapsed time from surgery to survey was 7.2±2.5 days.

One hundred forty-one patients had unknown addresses, including 131 males and 10 females. The mean age was 29.5±13.2 years. Eighty-four patients (59.6%) had a smoking history. The number of postoperative complications was 4 (2.8%), the number of ipsilateral recurrences was 3 (2.1%), and the number of contralateral recurrences was 1 (0.7%). Ninety-four (66.7%) of the patients lived in the city, and 47 (33.3%) lived outside the city. The mean postoperative hospital stay was 3.5±3.2 days, and the mean elapsed time from surgery to the survey was 8.4±2.0 days.

Comparisons among the three groups revealed significant differences in age (P = 0.0008), sex (P = 0.01), number of postoperative contralateral cases (<0.0001), place of residence at the time of surgery (P = 0.0002), and time between surgery and survey (<0.0001).

## Search for factors affecting the survey

Factors influencing the survey were searched in univariate and multivariate analyses (Table 2). Univariate analysis of factors influencing replies was performed via logistic regression analysis. The results were as follows: age (odds ratio (OR): 1.02 95% confidence interval CI (95% CI): 1.01–1.03 P = 0.0002), male sex (OR: 0.44 95% CI: 0.25–0.78 P = 0.0045), smoking history (OR: 0.74 95% CI: 0.54–1.00 P = 0.0508), postoperative complications (OR: 1.55 95% CI: 0.76–3.16 P = 0.23), postoperative ipsilateral recurrence (OR: 1.96 95% CI: 0.80–4.80 P = 0.139), postoperative contralateral incidence (OR: 8.81 95% CI: 2.67–29.1 P = 0.0.0004), place of residence at the time of surgery (OR: 0.75 95% CI: 0.55–1.02 P = 0.066), hospital stay after surgery

**Table 2. Factors affecting replies.**

| | Univariate analysis | | | Multivariate analysis | | |
|---|---|---|---|---|---|---|
| | Odds ratio | 95%CI | P value | Odds ratio | 95%CI | P value |
| Age | 1.02 | 1.01–1.03 | 0.0002 | 1.02 | 1.01–1.03 | <0.0001 |
| Male | 0.44 | 0.25–0.78 | 0.0045 | 0.39 | 0.22–0.71 | 0.002 |
| Smoking history | 0.74 | 0.54–1.00 | 0.0508 | | | |
| Postoperative complications | 1.55 | 0.76–3.16 | 0.23 | | | |
| Postoperative ipsilateral recurrence | 1.96 | 0.80–4.80 | 0.139 | | | |
| Postoperative contralateral incidence | 8.81 | 2.67–29.10 | 0.000359 | 12.1 | 3.55–41.20 | <0.0001 |
| Place of residence at the time of surgery (local or outside the city) | 0.75 | 0.55–1.02 | 0.0655 | | | |
| Postoperative hospital days | 1.01 | 0.96–1.05 | 0.76 | | | |
| Time from surgery to survey | 0.85 | 0.80–0.91 | <0.0001 | 0.842 | 0.79–0.90 | <0.0001 |

Examination of factors expected to influence replies when replies are considered positive.

(OR: 1.01 95% CI: 0.96–1.05 P = 0.76), and time from surgery to study (OR: 0.85 95% CI: 0.80–0.91 P<0.

In addition, multivariate analyses of age, male sex, postoperative contralateral onset, and time from surgery was conducted as explanatory variables.

The results were as follows. Age (odds ratio: 1.02 95% CI: 1.01–1.03 P = 0.0004), male sex (OR: 0.37 95% CI: 0.21–0.67 P<0.0001), postoperative contralateral incidence (OR: 13.4 95% CI: 3.97–45.1 P<0.0001), and time from surgery to investigation (OR: 0.84 95% CI: 0.79–0.90 P<0.0001) were included.

### Postoperative ipsilateral recurrence rate and postoperative contralateral pneumothorax incidence rate

On the basis of the results of this survey, the postoperative ipsilateral recurrence rate and postoperative contralateral incidence rate were calculated from 1–5 years after surgery via the Kaplan-Meier method for each age group from teenagers to 60 years.

First, the postoperative ipsilateral recurrence rates are shown in the order of 1 year, 2 years, 3 years, 4 years, and 5 years postoperatively by age group:

- **Teens**: 3.2% (1.5–7.0), 4.3% (2.2–8.4), 4.8% (2.5–9.0), 4.8% (2.5–9.0), and 4.8% (2.5–9.0)

- **20 s**: 0% (NA-NA), 0.5% (0.1–3.7), 1.1% (0.3–4.2), 2.2% (0.8–5.6), 2.2% (0.8–5.6)

- **30 s**: 0.9% (0.1–6.5), 0.9% (0.1–6.5), 0.9% (0.1–6.5), 0.9% (0.1–6.5), 0.9% (0.1–6.5)

- **40 s**: 0% (NA-NA), 1.6% (0.2–10.6), 1.6% (0.2–10.6), 1.6% (0.2–10.6), and 1.6% (0.2–10.6)

- **50 s**: 0% (NA-NA), 0% (NA-NA), 0% (NA-NA), 0% (NA-NA), 0% (NA-NA)

- **60 s**: 0% (NA-NA), 1.3% (0.2–8.6), 1.3% (0.2–8.6), 1.3% (0.2–8.6), and 1.3% (0.2–8.6)

The incidence of postoperative contralateral pneumothorax is also shown:

- **Teens**: 6.4% (3.7–11.1), 10.7% (7.0–16.1), 11.8% (7.9–17.3), 11.8% (7.9–17.3), 11.8% (7.9–17.3)

- **20 s**: 1.6% (0.5–4.8), 2.6% (1.1–6.2), 3.2% (1.4–6.9), 3.2% (1.4–6.9), 3.2% (1.4–6.9)

- **30 s**: 0% (NA-NA), 0% (NA-NA), 0% (NA-NA), 0% (NA-NA), 0% (NA-NA)

- **40 s**: 0% (NA-NA), 0% (NA-NA), 0% (NA-NA), 0% (NA-NA), 0% (NA-NA)

- **50 s**: 0% (NA-NA), 0% (NA-NA), 0% (NA-NA), 0% (NA-NA), 0% (NA-NA)

- **60 s**: 0% (NA-NA), 0% (NA-NA), 0% (NA-NA), 0% (NA-NA), 0% (NA-NA)

Postoperative ipsilateral recurrence rates were significantly different (P = 0.028) only at 1 year after surgery. Postoperative contralateral recurrence was observed only in those in their 10s and those in their 20s, with a particularly high incidence in teenagers, whose incidence was significantly greater than that in other age groups.

### Discussion

Compared with malignant diseases such as lung cancer, SP has a low complication rate and a recurrence rate of approximately 10% [3, 4], making long-term management less imperative. Furthermore, individuals in late adolescent are most susceptible to SP, often moving out of the medical care area owing to pursuits such as education and employment, making long-term follow-up challenging owing to social circumstances.

Post-treatment outcomes of SP are often evaluated through individual surveys for each study. Methods include telephone interviews, mailed questionnaires, and digital surveys such as email. In recent years, the protection of personal information has become a priority in Japan, making telephone interviews without prior notification challenging. However, a survey conducted by Harris et al. [11] in 1997 revealed that telephone interviews yielded higher response rates and were more cost-effective than mailed surveys were. On the other hand, direct telephone interviews by researchers may not yield accurate results if the questions are difficult for the subjects to answer [10].

In this study, we chose to administer a written survey approved by our hospital's ethics committee, and the effective response rate was 70.7%, with 21.0% of the addresses being unknown. Among the 141 individuals with unknown addresses, 83.7% were young individuals under the age of 40 years, thereby highlighting the challenges of administering surveys to evaluate the prognosis of SP. Surveys designed to evaluate the prognosis of patients with SP are often limited to approximately two years after surgery, with fewer studies focusing on long-term outcomes. As revealed by the multivariate analysis in this study, the difficulty of conducting follow-up surveys increases with the number of years after surgery. As a solution, digital surveys, as created in Uhlig et al.'s [12] cohort study, may offer higher response rates and cost-effectiveness. Particularly for younger subjects in today's widespread internet and smartphone usage era, digital surveys may be more beneficial.

Next, we discuss the results obtained from this survey. Compared with other age groups, teenagers had a higher rate of ipsilateral recurrence after surgery, especially in the first year after surgery. The peak rates of ipsilateral recurrence and contralateral onset in teenagers were observed at three years after surgery. These findings differ from those of previous studies [1], suggesting the need for longer follow-up observations than previously reported. Based on the results of this study, follow-up of teenagers for three years postoperatively is recommended, although new challenges such as staffing and health care costs arise. Compared with those for other age groups, the ipsilateral recurrence rate was 2.2% and the contralateral onset rate was 3.2% five years after surgery for individuals in their twenties, and the incidence of pneumothorax in individuals aged 30 years and above was negligible as limited follow-up observations were potentially sufficient.

The results of the present study indicate that teenage SP patients are more likely to develop pneumothorax. We believe that effective treatment should be selected from the outset for patients in the age group most prone to SP. Surgery is currently the most effective option. For postoperative recurrence, the use of polyglycolic acid sheets and additional adhesive therapies such as pleural abrasion or pleurectomy have been used for a long time. However, these methods are associated with inflammation in the thoracic cavity, naturally leading to pleural adhesions. In the future, it is necessary to find a method to prevent recurrence that does not rely on adhesions. Current surgical methods alone do not provide adequate care for patients with contralateral pneumothorax, which has a 10% incidence on the non-operative side. We believe that continuous follow-up of patients with SP, whose pathogenesis is unknown, is necessary to accumulate as much data as possible.

## Conclusions

In this survey, the difficulty of evaluating long-term outcomes after surgery in young individuals, who are prone to SP, became apparent. While exploring new tools such as digital surveys may be necessary, conducting large-scale surveys such as this one is crucial for standardizing the treatment of SP. According to the results of this survey, teenagers presented higher rates of ipsilateral recurrence and contralateral onset after surgery, with this trend peaking at three

years postoperatively, indicating the need for longer follow-up observations than previously considered. On the other hand, for individuals in their twenties and older, due to the lower incidence of pneumothorax after surgery, conducting periodic follow-ups may be less cost-effective than conducting comprehensive prognosis surveys over a short period. In such cases, using tools other than paper-based methods, such as digital tools, may further increase response rates.

## Supporting information

**S1 File. Survey content complete.**
(DOCX)

## Acknowledgments

We would like to take this opportunity to thank Dr. Toshiharu Tabata for conducting this survey with us.

## Author Contributions

**Conceptualization:** Ryo Nonomura.

**Data curation:** Ryo Nonomura.

**Formal analysis:** Ryo Nonomura.

**Investigation:** Ryo Nonomura, Ryuga Yabe, Yutaka Oshima, Takanobu Sasaki, Naoya Ishiba-shi, Takafumi Sugawara.

**Methodology:** Ryo Nonomura.

**Project administration:** Ryo Nonomura.

**Writing – original draft:** Ryo Nonomura.

**Writing – review & editing:** Ryo Nonomura.

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
