## [Decision Letter · Decision Letter 0]

24 Jun 2024

PONE-D-24-20954Spontaneous Pneumothorax Post-Surgery: Long-Term Recurrence Rates and Follow-Up Challenges Revealed by a Written SurveyPLOS ONE

Dear Dr. Nonomura,

Thank you for submitting your manuscript to PLOS ONE. After careful consideration, we feel that it has merit but does not fully meet PLOS ONE’s publication criteria as it currently stands. Therefore, we invite you to submit a revised version of the manuscript that addresses the points raised during the review process.

We look forward to receiving your revised manuscript.

Kind regards,

Luca Bertolaccini, M.D., Ph.D.

Academic Editor

PLOS ONE

Journal Requirements:

5. Please include your tables as part of your main manuscript and remove the individual files. Please note that supplementary tables (should remain/ be uploaded) as separate "supporting information" files

Additional Editor Comments:

The reviewers have emphasised issues that require a careful and thorough revision of the manuscript.

No commitment to publication can be made at this point.

Reviewers' comments:

Reviewer's Responses to Questions

**Comments to the Author**

1. Is the manuscript technically sound, and do the data support the conclusions?

Reviewer #1: Yes

Reviewer #2: Partly

2. Has the statistical analysis been performed appropriately and rigorously? 

Reviewer #1: Yes

Reviewer #2: Yes

3. Have the authors made all data underlying the findings in their manuscript fully available?

Reviewer #1: Yes

Reviewer #2: Yes

4. Is the manuscript presented in an intelligible fashion and written in standard English?

Reviewer #1: Yes

Reviewer #2: Yes

5. Review Comments to the Author

Reviewer #1: Dear Authors,

I wanted to express my appreciation for your manuscript titled "ISpontaneous Pneumothorax Post-Surgery: Long-Term Recurrence Rates and Follow-Up Challenges Revealed by a Written Survey". Your work demonstrates a commendable level of scholarship, presenting original and insightful research that significantly contributes to the field of thoracic surgery.

However, as I delved into your manuscript, I noticed a few areas where revisions could enhance its clarity and impact. Firstly, I suggest uploading the survey as a supplementary material.

There are minor aspects to improve then. In the introduction the SP is directly written as an abbreviation. There are few phrases where the construction is strange (e.i: …In Japan, since the enactment of the Personal 91 Information Protection Law Harris LE et al. 1997 found that a telephone 92 interview survey was more cost-effective than a mailed written survey because of 93 the higher response rate….).

Discussion may be improved. There should be a discussion on treatment, recurrence prevention etc.

References may be updated. e.i. Walker S, Hallifax R, Ricciardi S, et al. Joint ERS/EACTS/ESTS clinical practice guidelines on adults with spontaneous pneumothorax. Eur Respir J. 2024;63(5):2300797. Published 2024 May 28. doi:10.1183/13993003.00797-2023

In conclusion, I believe that with these revisions, your manuscript will make an even more significant contribution to our understanding of thoracic surgery. I commend you for your diligent work and eagerly anticipate seeing the revised manuscript.

Thank you for your attention to these suggestions

Reviewer #2: I congratulate the authors for their efforts. Conscise and well written paperIt was conducted as a cross-sectional survey study,

At this point, I have some critic:

-Why wasn't the study conducted only among those who gave clear answers?

-What is the reason why the postoperative ipsilateral recurrence rates are in the 50s and the postoperative contralateral pneumothorax rates are 0% in the 30-60s?

-Again, depending on this, the fact that the bleps are in different lobes in SP cases and the pleurodesis method used may affect it (pleural abrasion, pleurectomy, chemical pleuredesis). Different pleurodesis methods may have been used in different age groups. What are the authors' comments on this subject?

6. PLOS authors have the option to publish the peer review history of their article (what does this mean?). If published, this will include your full peer review and any attached files.

Reviewer #1: **Yes: **Alessio Campisi

Reviewer #2: **Yes: **yekta altemur karamustafaoglu

---

## [Author Response · Author response to Decision Letter 0]

9 Jul 2024

Thank you for reviewing my paper.

I have added and revised the parts you pointed out.

The details are described in the file "to reviewer".

If you have any additional comments or suggestions, please let me know and I will correct them.

---

## [Decision Letter · Decision Letter 1]

15 Jul 2024

Spontaneous Pneumothorax Post-Surgery: Long-Term Recurrence Rates and Follow-Up Challenges Revealed by a Written Survey

PONE-D-24-20954R1

Dear Dr. Nonomura,

We’re pleased to inform you that your manuscript has been judged scientifically suitable for publication and will be formally accepted for publication once it meets all outstanding technical requirements.

Kind regards,

Luca Bertolaccini, M.D., Ph.D.

Academic Editor

PLOS ONE

Additional Editor Comments (optional):

Reviewers' comments:

Reviewer's Responses to Questions

**Comments to the Author**

1. If the authors have adequately addressed your comments raised in a previous round of review and you feel that this manuscript is now acceptable for publication, you may indicate that here to bypass the “Comments to the Author” section, enter your conflict of interest statement in the “Confidential to Editor” section, and submit your "Accept" recommendation.

Reviewer #1: All comments have been addressed

Reviewer #2: All comments have been addressed

2. Is the manuscript technically sound, and do the data support the conclusions?

Reviewer #1: Yes

Reviewer #2: Yes

3. Has the statistical analysis been performed appropriately and rigorously? 

Reviewer #1: Yes

Reviewer #2: Yes

4. Have the authors made all data underlying the findings in their manuscript fully available?

Reviewer #1: Yes

Reviewer #2: Yes

5. Is the manuscript presented in an intelligible fashion and written in standard English?

Reviewer #1: Yes

Reviewer #2: Yes

6. Review Comments to the Author

Reviewer #1: Dear Authors,

I have reviewed your revised manuscript and am pleased to inform you that all the comments and suggestions raised during the initial review have been adequately addressed. The revisions have significantly improved the quality of the paper. I am recommending it for publication.

Best regards,

Alessio Campisi

Reviewer #2: Thanks you for your corrections and answers.I believe

that these revisions address the concerns raised by the other reviewers and the managing editor.

7. PLOS authors have the option to publish the peer review history of their article (what does this mean?). If published, this will include your full peer review and any attached files.

Reviewer #1: **Yes: **Alessio Campisi

Reviewer #2: **Yes: **yekta altemur karamustafaoglu

---

## [Editor Report · Acceptance letter]

30 Sep 2024

PONE-D-24-20954R1 

PLOS ONE

Dear Dr. Nonomura, 

I'm pleased to inform you that your manuscript has been deemed suitable for publication in PLOS ONE. Congratulations! Your manuscript is now being handed over to our production team.

Kind regards, 

on behalf of

Dr. Luca Bertolaccini 

Academic Editor

PLOS ONE